# Sustainable Fashion in New Era: Exploring Consumer Resilience and Goals in the Post-Pandemic

Joohye Hwang [1], Xun Sun [2], Li Zhao [2] and Song-yi Youn [2,*]

1 School of Business, Thomas Jefferson University, Philadelphia, PA 19144, USA; joohye.hwang@jefferson.edu
2 Textile and Apparel Management, University of Missouri, Columbia, MO 65211, USA; xshmt@mail.missouri.edu (X.S.); zhaol1@missouri.edu (L.Z.)
* Correspondence: syoun@missouri.edu

**Abstract:** This study examines the underlying mechanisms that lead to sustainable fashion consumption in the post-COVID-19 pandemic era. Particularly, this study explores the complex relations between resilient coping mechanisms, consumer life goals, and sustainable fashion consumption, combining Goal Content Theory and the Consumer Sustainability Orientation framework. The findings obtained from partial least squares structural equation modeling analysis using 503 responses confirm that resilient coping positively influenced both intrinsic and extrinsic life goals. While intrinsic goals reinforce all aspects of sustainability orientation (ecological, social, and economic), extrinsic goals show a contrasting effect—positively affecting economic orientation but negatively impacting ecological and social dimensions. Among sustainability orientations, the ecological dimension had a significant positive effect on sustainable fashion consumption behavior. The research also reveals that resilient coping directly improves social and economic dimensions of sustainability orientations, but it does not significantly influence ecological orientation. This study offers insight into differentiated marketing communication strategies for retailers depending on consumers' goals—intrinsic or extrinsic—and implies the importance of the dynamic impact of each dimension of sustainability orientation on consumers' sustainable fashion consumption choices.

**Keywords:** sustainable fashion; resilient coping; intrinsic goals; extrinsic goals; ecological orientation; social orientation; economic orientation; post-pandemic

## 1. Introduction

The way people approach their lives and make decisions as individual consumers has been transformed fundamentally by COVID-19. This shift is evident through disruptions in subsequent economic recovery and consumption behaviors [1,2]. For the fashion industry, the post-pandemic environment has posed serious obstacles. For instance, an extensive amount of instability has affected fashion companies such as H&M, resulting in the closure of almost 70% of its locations worldwide [3,4]. Moreover, concerns related to sustainability have been raised, such as the working conditions in warehouses of online fashion brands (e.g., ASOS, Shein (ultra-fast fashion brand), etc.; [3]. Particularly, in the intricate fashion industry, environmental concerns and ecological sustainability awareness have significantly increased during the pandemic due to severe disruptions in the fashion supply chain [5]. Consequently, consumers have become more conscious of the environmental impacts of their fashion consumption choices. They have adopted sustainable methods of acquiring and consuming fashion products while increasing support for fashion retailers implementing environmentally friendly business practices [6]. This shift reflects consumers' recognition of the intimate connection between fashion and their daily lives. Thus, the fashion industry has faced a need for rapid restructuring across its supply chain to meet consumer demands for sustainability [7].

Changes in consumer behavior triggered by COVID-19 have been addressed within two research streams. The first stream has focused on a noticeable trend toward the

direction of sustainable consumption, which involves an increase in demand preference for eco-friendly and ethically sourced fashion products [8,9]. This trend entails the adoption of various forms of sustainable practices such as reuse, recycling, slow fashion, and cruelty-free production throughout the fashion supply chain [2,10,11]. The other research stream has focused on how consumers have responded to challenges caused by the pandemic (e.g., social isolation and depression; [8,12–14]). The challenges prompted consumers to reassess their priorities and values, including reconsidering perceived attitudes toward risks and new social norms and expectations of shopping [15], ultimately affecting their consumption choices and shopping behavior (e.g., channel switching to online; [16]).

Although these two research streams have been investigated concurrently, there is a limited approach to connecting sustainable fashion consumption movements and consumers' psychological responses in the post-pandemic context. According to several previous studies, consumers have learned how to navigate this pandemic-related stress and frustration and developed resilient coping skills to recover themselves from adverse experiences and overcome difficulties (e.g., emotions, resilience, and optimism; [12,14,17–19]). By extending previous findings, this study aimed to explore the interrelations between coping mechanisms and sustainable consumption behavior by identifying underlying factors that affect these relationships.

To achieve the goal, this study first adopted Goal Content Theory (GCT) to understand factors that mediate between resilient coping skills and sustainable fashion consumption. GCT provides a detailed comprehension of intrinsic and extrinsic goal motivations [20], factors known to impact sustainable behaviors. Previous scholars have focused on the environmental side of sustainable behavior and investigated the connection between intrinsic goal motives and eco-friendly consumption behavior, and some research has emphasized how environmentally friendly consumption influences consumers' life satisfaction [21–23]. This study extended these to fashion consumption in the COVID-19 pandemic context to understand the mechanisms that explain the pandemic impact, the role of resilience, consumers' goals, and their subsequent sustainable behavior changes.

Moreover, concerning the relationship between consumer goals and sustainable fashion consumption, this study aimed to build a theoretical framework that explains how consumer goals—intrinsic and extrinsic goals—affected by the pandemic shape consumers' sustainability orientation toward fashion consumption by adopting the Consumer Sustainability Orientation (CSO) framework. It explains three sub-dimensions—i.e., ecological, social, and economic orientation—and provides a comprehensive lens to understand how the pandemic has influenced consumers' goals and their sustainability orientations [24,25].

Finally, to include behavioral consequences in the research discussion, this study adopted the concept of sustainable fashion consumption (SFC), which refers to consumers' consumption practices that prioritize environmentally friendly sourced and constructed fashion products, as well as seeking longer use of fashion items [26]. SFC has been discussed in previous studies, particularly in association with environmental impacts and consumers' shopping behavior [26]. In line with this, this study explored not only the relationships between consumers' psychological mechanisms and multi-faceted sustainable orientations but also their sustainable fashion consumption behavior, with a specific focus on environmental considerations within the COVID-19 pandemic-induced context.

Therefore, by incorporating GCT into the CSO framework and adopting SFC, this study sought to provide a comprehensive insight into understanding the dynamic relations between psychological resilience, consumers' goals, sustainable orientations, and their sustainable consumptions in the context of fashion, particularly in the post-pandemic era. The following literature review section was constructed in the following order: firstly, this study explored how resilient coping with the COVID-19 pandemic has influenced consumers' intrinsic and extrinsic goals; secondly, it examined how these consumers' goals have affected their sustainability orientations; and thirdly, how consumers' sustainability orientations have driven their sustainable fashion consumption behaviors.

## 2. Literature Review

### 2.1. Resilient Coping under Post-Pandemic

According to previous studies in psychology, resilience coping refers to the ability to rebound from difficult and challenging events or experiences in life [27,28]. In a disastrous situation, it explains how individuals can positively adapt to protect themselves from becoming distressed by managing stressful situations and recovering from cognitive or social functional disorders [27,29–31]. In contrast to negative emotions (e.g., anxiety, fear, etc.) evoked as a response to traumatic events, resilient coping is a positive psychological response to manage mental distress [29]. Some researchers have focused on identifying factors that induce different patterns of resilience. For instance, personality traits and individual characteristics have been explored as factors that differentiate the level of resilience and coping behavior among individuals [17]. In line with this, multidimensional resilience (i.e., active and passive resilience) has been discussed along with individuals' emotional experiences. While active resilience is a more proactive adjustment to disruptions, passive resilience focuses on the ability to endure stressful events [17,32]. Other researchers have shifted their point of view and expanded their focus to the protective process of resilient coping or the outcome, such as life well-being, personal growth, and performance improvement [33–35].

In particular, resilient coping has gained increased attention in the consumer study field within the context of the COVID-19 pandemic as researchers seek to understand individuals' psychological mechanisms to overcome additional challenges that have arisen due to the pandemic [8,36,37]. The existing literature has mainly discussed changes in consumer consumption behavior by exploring consumers' value shifts, resilience, and stress coping. During the pandemic, various consumption practices were identified, specifically focusing on clothing and apparel consumption [8,36,38]. For example, Liu et al. [36] explored several themes, including safety concerns, backlogged demand, and consumption transition (e.g., dieting their closet and donations to the community) by analyzing tweets. Scholars found that clothing consumption change is a significant resilience process to deal with pandemic distress as it gives consumers control in their difficult, uncertain, and ambiguous times [8,36,37]. Kursan Milaković [39] also examined the role of consumer resilience on consumption satisfaction and repurchase intention, along with the variables of consumer vulnerability and adaptability. The researcher conducted a quantitative analysis to investigate consumers' coping capacities during the pandemic and confirmed that consumer resilience increased consumer repurchase intention mediated through satisfaction [39].

### 2.2. Goal Content Theory (GCT): Intrinsic and Extrinsic Goals

This study adopted Goal Content Theory (GCT) as a main theoretical framework. GCT is a mini-theory of self-determination theory (SDT), a widely accepted theory in the field of psychology that explains human motivation, personality, growth, and well-being [40]. The theory originated from the discovery of the undermining effects of intrinsic motivation and has been extended to a larger theory with several sub-frameworks [40]. These researchers found that humans have a true intrinsic motivation, which is they obtain pleasure from simply doing something; if external rewards or punishments are provided, intrinsic motivation diminishes. Anchored by intrinsic motivation, the notion of "self" that has an active integrative nature was defined, and SDT was developed [40]. By reflecting the organismic perspective [41], SDT defined "self" as the subjective "I" that regulates behavior and coordinates external and internal inputs, which leads to continuous personal growth [40]. The effective self-functioning process brought attention to the fundamental psychological needs (i.e., autonomy, competence, and relatedness) as requirements for humans to thrive and experience psychological well-being.

Transitioning from this foundational understanding of SDT, the focus then shifted to how specific motivations shape behavior [40]. This shift led to Goal Content Theory (GCT) within the SDT framework. According to GCT, goal content refers to the actual objectives or ends that people strive to achieve [40]. It categorizes goals into two types: intrinsic

and extrinsic goal content. Intrinsic goals include personal growth, emotional intimacy, and community involvement, which are not related to external rewards and goals that can inherently be rewarding to individuals. Conversely, extrinsic goals refer to wealth, financial success, popularity/recognition, and appearance/image, which are externally oriented contents [40,42].

GCT scholars explain the "what" of individual motivations (i.e., internal or external goals), which is a different approach than focusing on the "why" of underlying motivations driving individuals toward their goals [20,43]. The approaches differentiate between the goal attainment process and the nature of the goals themselves, underscoring the influence of goal content on psychological health and performance. The researchers found that when the goal contents were more extrinsic rather than intrinsic, people expected to be engaged in less happy feelings, and those who pursued more intrinsic goals rather than extrinsic goals showed positive psychological effects [43]. Also, the negative relationships between extrinsic goal content and subsequent psychological reactions have been confirmed [20].

In the consumption context, previous scholars explained the relationship between fashion consumers' intrinsic and/or extrinsic goals and their subsequent shopping-related behaviors [44,45]. For example, consumers who are motivated more by extrinsic orientations (i.e., image enhancement, popularity, or financial achievement) over intrinsic goals (i.e., self-development, community contribution, relationship building, or sustainability) tend to prefer products and brands that show their social status [44]. Additionally, they are more likely to use brand names as a means to express their self-identity, especially in their interactions and presentations on social media [45].

While there is a significant body of literature on motivational factors influencing sustainable consumption practices in consumer studies, limited research examines the motivational mechanism of sustainable consumption by employing resilient coping in navigating sustainable consumption through intrinsic and extrinsic goals. Understanding the psychological mechanism and interactions among motivational determinants can shed light on the complexities of sustainable consumption decision-making processes and elucidate the dynamic process of individuals drawing upon resilient coping strategies to pursue sustainable fashion consumption [46].

*2.3. The Influence of Resilient Coping on Intrinsic and Extrinsic Goals*

Resilient coping and goal motives (i.e., intrinsic and extrinsic) in the context of pandemic distress situations have been discussed in the previous literature [18]. While navigating through the pandemic, consumers became more oriented to extrinsic goals to cope with their mental distress caused by a lack of physical resources. The pandemic caused many businesses to close due to the lockdown, and a shortage of materials also adversely influenced the retail industry. As a result, consumers experienced a shortage of resources, which caused them to have a sense of resource scarcity, which might have motivated them to purchase more products and prioritize extrinsic goals [14]. In other words, the pandemic caused consumers psychological distress, and they focused more on extrinsic goals as compensation [14]. Simultaneously, consumers' resilient coping may enhance their intrinsic goals by helping them develop skills to manage challenging situations (e.g., the COVID-19 pandemic; [20]), which emphasizes the importance of self, meaningful life, and building connections with others and caring for communities [47]. Thus, we proposed the following:

**H1.** *Consumers' resilient coping with COVID-19 will positively affect their intrinsic goals.*

**H2.** *Consumers' resilient coping with COVID-19 will positively affect their extrinsic goals.*

*2.4. Consumer Sustainability Orientation Framework*

This study adopted the Consumer Sustainability Orientation (CSO) framework to actualize application to sustainable fashion consumption in the research model. It extends

the triple-bottom-line theory, describes how consumers value the three facets of sustainability, including ecological, social, and economic, and integrate them into their consumption practices [23,48]. According to the framework, sustainable consumption orientation refers to consumers' overall attitudes toward companies, products, and (business) practices that have environmental, social, and economic impacts on their lives, not only in the short-term but also in the long run [23]. Triple-bottom-line sustainability was initially suggested to assess business performance based on three aspects, planet, people, and profits, and further developed into environmental, social, and economic sustainability [49]. The multi-faceted triple-bottom-line framework has facilitated businesses to expand beyond their environmental awareness and promoted them to include social and economic aspects for their sustainable success [25,46,50].

According to the CSO framework, sustainable orientations reflect the extent to which individuals feel responsible for and prioritize sustainable practices in their consumption choices [23,51]. The framework covers a wide range of consumer behaviors and attitudes by integrating environmental concern, and economic and social responsibility aspects [23], and it provides a comprehensive understanding of sustainable consumption, particularly focusing on consumers' purchasing decisions and lifestyle choices [25,51]. Within the framework, ecological orientation is related to the business practices that ensure the preservation of natural resources for future generations. Social orientation relates to business practices that value the well-being of people, communities, and society the most. Economic orientation refers to business practices that consider the economic contributions of organizations to society [25,48]. Based on these three dimensions, consumers' perspectives on sustainability efforts have been investigated by focusing on consumer values and their attitudes toward businesses' sustainable activities. For example, previous scholars pointed out the lack of attention to social and economic dimensions of sustainability orientation within consumer studies and emphasized the importance of a balanced perspective to examine consumers' sustainability orientation considering the three sub-dimensions—ecological, social, and economic [52]. In the context of fashion retail, sustainability orientations have been examined as critical determinants of environmentally and socially responsible apparel consumption behavior, and the discussion considers varying levels of sensitivity toward sustainability based on the consumer generation for effective communication with the target consumers (e.g., green consumption values and environmental consciousness; [53,54]).

While previous studies have identified a variety of antecedents contributing to individuals' sustainable orientation, such as personal characteristics, social demographics, and value orientation [55,56], the attitude–behavior gap in sustainable consumption often arises from individual cognitive factors, including consumers' knowledge about brands and brand image shaped by business practices [56]. Therefore, by integrating sustainability orientation and examining the psychological motivational factors behind sustainable fashion consumption, we can gain further insights into understanding how these factors shape decisions regarding sustainable fashion consumption.

### 2.5. The Effect of Consumer Goals on Sustainability Orientation

In the context of the pandemic, individuals have demonstrated awareness and concern for social and environmental sustainability issues [57]. Scholars have argued that various psychological factors, such as personal values and motivation [58] and self-identity [59], are in relation to determinants of sustainability orientation [60]. Previous studies have emphasized consumers' goal motivation toward consumers' tendency to engage in sustainability-related behaviors [23,51]. According to Buerke et al. [58], intrinsic goals value the relationship between the self and others/the environment, enabling consumers to feel a sense of societal benevolence. These goals are posited to positively affect fashion consumers' ecological and social sustainability orientations. On the other hand, extrinsic goals, associated with material values, are expected to negatively affect fashion consumers' ecological and social orientations while boosting their economic orientation [51,52]. Therefore, we proposed the following hypotheses:

**H3.** *Consumers' intrinsic goals will positively affect their (a) ecological, (b) social, and (c) economic sustainable orientation.*

**H4.** *Consumers' extrinsic goals will negatively affect their (a) ecological and (b) social sustainable orientations but will positively affect (c) economic sustainable orientation.*

*2.6. The Direct Effect of Resilient Coping on Consumers' Sustainability Orientation*

Previous studies have examined the interplay relationship between resilience and sustainability across multiple fields via diverse lenses. For example, Etse et al. [61] investigated disaster resilience and urban sustainability and highlighted the significance of resilience as a necessity for urban system sustainability. Dhir et al. [62] examined individuals' overall sense of positive expectations toward a certain event and their internal feelings and motivation toward organizations' sustainable labeling behaviors. In the context of the COVID-19 pandemic, in order to cope with the emotional and psychosocial stress from COVID, Shanahan et al.'s [37] study showed that coping strategies such as positive reappraisal/reframing could mitigate distress levels. In particular, the concept of individual resilience coping has been widely recognized as a positive and flexible adaptability in the face of risk; specifically, people who possess ego-resilience demonstrate greater self-transcendent personal goals [27,35]. In addition, sustainable consumption orientation reflects consumers' value of sustainability, leading to positive change in the environment, society, and economy. Consumers with such positive coping strategies are likely to engage in a constructive reframing of their consumption perspectives [36], aligning with sustainability values. Thus, we proposed the following hypothesis:

**H5.** *Resilient coping will have a positive direct effect on fashion consumers' (a) ecological, (b) social, and (c) economic sustainable orientations.*

*2.7. The Effect of Consumers' Sustainability Orientation on Their Sustainable Fashion Consumption*

Consumers' sustainable consumption behavior is motivated by various and complex factors, particularly the role of values that have been studied in the previous literature [63]. Consumer sustainability orientation describes the consumers' personal value toward sustainability, encompassing considerations of beliefs, ethics, and moral perspectives on sustainability issues [63]. This orientation is particularly evident in the aspects of natural preservation, social justice, individual's well-being, and organizational sustainable development [52]. Prior studies found that consumer awareness and sustainability-focused value orientation are the central psychological antecedents of consumers' responsible consumption behavior [58]. These studies primarily identified that self-values such as morality and responsibility are commonly linked with more sustainable consumption [64]. Dhir et al. [62] found that transparency in sustainable consumption behaviors significantly influences consumer choices, aligning with ecological, social, and economic sustainability orientation. This reinforces consumers' sustainable consumption behaviors. These findings suggest that a clear sustainability mindset directly encourages consumers to opt for sustainable fashion, supporting the following hypothesis:

**H6.** *Fashion consumers' (a) ecological, (b) social, and (c) economic sustainable orientations will positively affect their sustainable fashion consumption.*

Based on the proposed relationships between the concepts explored in this study, a research model was established, as shown in Figure 1.

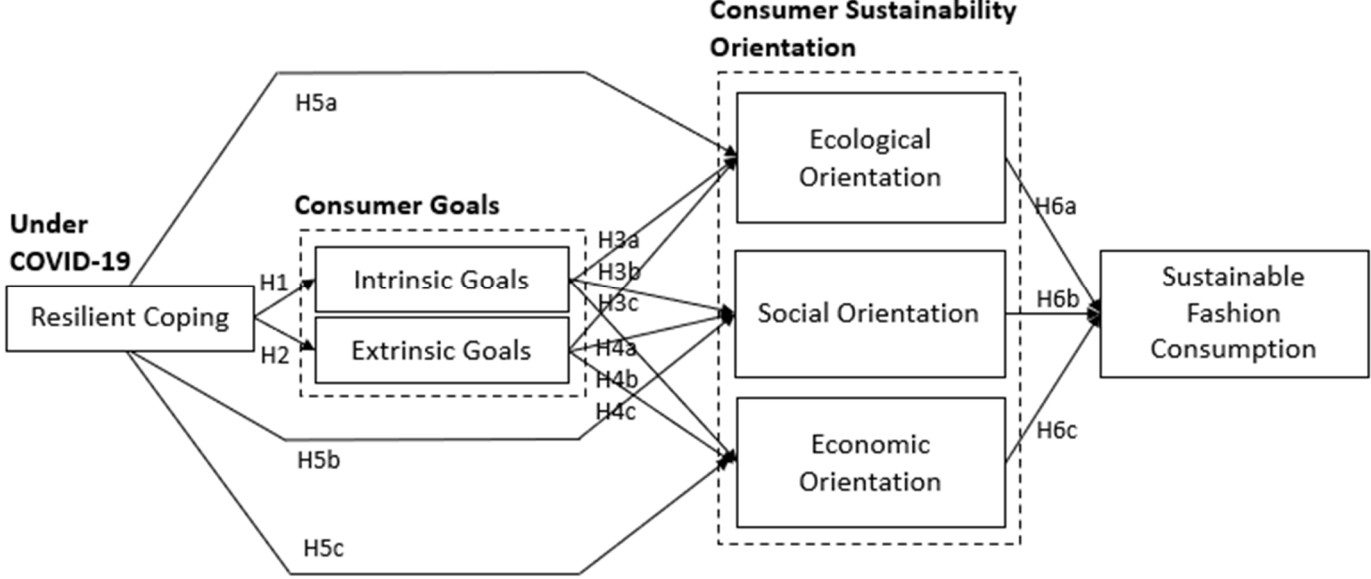

**Figure 1.** Research model.

### 3. Method

*3.1. Data Collection*

Before collecting the data, this study obtained approval from the Institutional Review Board (IRB). The researchers collected the data from 27 November to 30 November 2021, via Amazon Mechanical Turk (MTurk). The survey was online, and the participants were consumers 18 years or older living in the U.S. There were no screening questions except for age and location, as the survey asked consumers' thoughts about their resilience during COVID-19 and their future intention for sustainable fashion consumption behavior. The survey was designed to provide the context of fashion aligning with the study purpose as follows: firstly, participants were introduced to the changes in the fashion industry during the COVID-19 pandemic at the beginning of the survey; secondly, they were prompted to review and reflect on their fashion consumption behavior before answering the survey questionnaire. An amount of 0.7 dollars was paid to each participant, and Mturk provided survey participants compensation according to its standard reward policy. Unusable data were excluded using the listwise deletion approach, and further analysis was conducted based on a subset of complete cases (N = 503). Approximately 52% of the participants were female, and 46% were male. Most participants were from 25 to 44 years old (55%) and Caucasian (76%), followed by Asians (9%), African Americans (7%), Hispanic/Latinos (5%), and others (3%) (See Table 1).

**Table 1.** Demographic characteristics of the sample population (N = 503).

| Variable | Descriptions | Frequency | Percent (%) |
|---|---|---|---|
| **Gender** | Female | 262 | 52.1% |
| | Male | 232 | 46.1% |
| | Non-binary/third gender | 1 | 0.2% |
| | Prefer not to say | 8 | 1.6% |
| **Age** | 18–24 | 13 | 2.6% |
| | 25–29 | 60 | 11.9% |
| | 30–34 | 97 | 19.3% |
| | 35–39 | 67 | 13.3% |
| | 40–44 | 52 | 10.3% |

**Table 1.** *Cont.*

| Variable | Descriptions | Frequency | Percent (%) |
|---|---|---|---|
| | 45–49 | 35 | 7.0% |
| | 50–54 | 43 | 8.5% |
| | 55–59 | 47 | 9.3% |
| | 60–64 | 46 | 9.1% |
| | 65 or older | 43 | 8.5% |
| **Ethnicity** | Asian | 43 | 8.5% |
| | Black or African American | 35 | 7.0% |
| | Caucasian | 381 | 75.7% |
| | Hispanic/Latino | 25 | 5.0% |
| | Mixed Race | 14 | 2.8% |
| | Native American | 3 | 0.6% |
| | Others | 1 | 0.2% |
| | Pacific Islander | 1 | 0.2% |
| **Annual Household Income** | Less than $24,999 | 98 | 19.5% |
| | $25,000 to $49,999 | 147 | 29.2% |
| | $50,000 to $74,999 | 121 | 24.1% |
| | $75,000 to $99,999 | 82 | 16.3% |
| | $100,000 to $149,999 | 29 | 5.8% |
| | $150,000 to $199,999 | 11 | 2.2% |
| | $200,000 or more | 15 | 3.0% |
| **Employment** | Employed (part-time, full-time, self-employed) | 419 | 83.3% |
| | Student | 9 | 1.8% |
| | Unemployed or retired | 65 | 12.9% |
| | I prefer not to answer | 10 | 2.0% |
| **Education** | Less than high school | 3 | 0.6% |
| | High school | 51 | 10.1% |
| | Some college/college diploma/certificate | 111 | 22.1% |
| | Bachelor's degree | 227 | 45.1% |
| | Master's degree | 92 | 18.3% |
| | Doctorate | 13 | 2.6% |
| | I prefer not to answer | 6 | 1.2% |

*3.2. Measurement*

This study measured participants' resilient coping, consumption goals, sustainable orientation, and subsequent future sustainable fashion consumption behavior. All measurement items were adopted from the previous literature. Four items that asked about participants' resilient coping under COVID-19 were adapted from the Brief Resilient Coping Scale (BRCS; $\alpha = 0.82$; [12,31]). Six items to measure consumer life goals—intrinsic goals ($\alpha = 0.70$) and extrinsic goals ($\alpha = 0.74$)—were also adopted from the previous literature [20]. Consumer sustainability orientation—ecological orientation ($\alpha = 0.93$), social orientation ($\alpha = 0.93$), and economic orientation ($\alpha = 0.86$)—were adapted from [52]. Sustainable fashion consumption ($\alpha = 0.91$), which specifically focuses on the ecological aspect of sustainability, was adopted from Razzaq et al. [26] (see Table 2). All items were measured using a 5-point Likert scale. Resilient coping was measured by asking participants to indicate how well each statement describes their thoughts, ranging from "not well at all" to "extremely well." Consumer sustainability orientations (e.g., ecological, social, and economic) and sustainable fashion consumption were measured on a scale from "strongly disagree" to "strongly agree." Intrinsic and extrinsic goals were assessed by prompting participants to prioritize consumer goals when shopping for fashion products in the survey.

Participants were asked to indicate how well each statement reflects their consumer goals, ranging from "not well at all" to "extremely well."

**Table 2.** Assessment of measurement model.

| Construct | Loading | CR | AVE |
|---|---|---|---|
| **Resilient Coping (RES)** | | 0.879 | 0.646 |
| RES1: I look for creative ways to alter difficult situations. | 0.830 | | |
| RES2: Regardless of what happens to me, I believe I can control my reaction to it. | 0.731 | | |
| RES3: I believe I can grow in positive ways by dealing with difficult situations. | 0.827 | | |
| RES4: I actively look for ways to replace the losses I encounter in life. | 0.823 | | |
| **Intrinsic Goals (INT)** | | 0.821 | 0.604 |
| INT1: Self-acceptance/personal growth: Being happy and having a very meaningful life. | 0.750 | | |
| INT2: Intimacy/friendship: Having many close and caring relationships with others. | 0.771 | | |
| INT3: Societal contribution: Working to help make the world a better place. | 0.810 | | |
| **Extrinsic Goals (EXT)** | | 0.849 | 0.653 |
| EXT1: Physical appearance: Looking good and being attractive to others. | 0.793 | | |
| EXT2: Popularity/recognition: Being known and/or admired by many people. | 0.877 | | |
| EXT3: Financial success: Having a job that pays very well and having a lot of nice possessions. | 0.751 | | |
| **Ecological Orientation (ECOL)** | | 0.943 | 0.735 |
| ECOL1: It is important to me to take care of our environment. | 0.890 | | |
| ECOL2: It is important to me that the manufacturing of products does not harm our environment. | 0.884 | | |
| ECOL3: I think it is important that products can be recycled. | 0.802 | | |
| ECOL4: The long-run preservation of natural resources concerns me. | 0.844 | | |
| ECOL5: It is important to me that products are reusable to conserve natural resources. | 0.880 | | |
| ECOL6: It is important to me that companies reduce their emissions. | 0.841 | | |
| **Social Orientation (SOC)** | | 0.944 | 0.773 |
| SOC1: It is important to me that companies treat their employees fairly. | 0.876 | | |
| SOC2: It is important to me that the manufacturing of products does not conflict with human rights. | 0.898 | | |
| SOC3: It is important to me that companies act as a fair player in the marketplace. | 0.855 | | |
| SOC4: It is important to me that the manufacturing of products does not exploit the labor. | 0.900 | | |
| SOC5: I care about adequate wages for the workforce. | 0.865 | | |
| **Economic Orientation (ECON)** | | 0.916 | 0.784 |
| ECON1: It is important to me that companies are successful in the long run. | 0.900 | | |
| ECON2: It is important to me that companies gain adequate profits to survive in the market. | 0.903 | | |
| ECON3: It is important to me that companies are future-oriented. | 0.853 | | |
| **Sustainable Fashion Consumption (SFC)** | | 0.933 | 0.736 |
| SFC1: I will buy fashion product that is made with recycled content. | 0.870 | | |
| SFC2: I will buy fashion product that can be disposed of in an environmentally friendly manner. | 0.860 | | |
| SFC3: I will buy fashion product that is safe to the environment. | 0.873 | | |
| SFC4: I will limit my use of the fashion product that is made of or uses scarce resources. | 0.795 | | |
| SFC5: I will buy fashion product that is produced in an environmentally friendly manner. | 0.887 | | |

## 4. Results

### 4.1. Measurement Model Results

This study analyzed the survey results using the partial least squares (PLS) model. To evaluate the measurement model before conducting the path analysis, the measurements' reliability and the variables' discriminant validity were checked. The factor loadings of all items that were used for further analysis were $\geq 0.73$, and the composite reliability for all constructs was $\geq 0.82$, which indicated that all constructs were reliable. As for

the discriminant validity, both the Fornell–Larcker criterion and Heterotrait–Monotrait ratio were examined. The Fornell–Larcker criterion confirmed that this study met the discriminant validity requirement as the AVEs of the constructs were above 0.60, and the square root of the AVE was higher than the correlation between the two constructs [65] (see Table 3). Also, the results of the Heterotrait–Monotrait (HTMT) ratio of the correction technique did not exceed the threshold value of 0.90 [66] (see Table 4). This study also examined the variance inflation factors (VIFs) to detect any common method bias in the PLS-SEM. However, the VIF scores were all found to be below the threshold value of 3, indicating that the measurement model was free from common bias [66] (see Table 5).

**Table 3.** Assessment of discriminant validity: Fornell–Larcker criterion.

|  | **RES** | **INT** | **EXT** | **ECOL** | **SOC** | **ECON** | **SFC** |
|---|---|---|---|---|---|---|---|
| RES | 0.804 | | | | | | |
| INT | 0.479 | 0.777 | | | | | |
| EXT | 0.352 | 0.442 | 0.808 | | | | |
| ECOL | 0.244 | 0.393 | 0.075 | 0.857 | | | |
| SOC | 0.193 | 0.318 | −0.063 | 0.659 | 0.879 | | |
| ECON | 0.306 | 0.286 | 0.280 | 0.303 | 0.224 | 0.885 | |
| SFC | 0.216 | 0.367 | 0.093 | 0.787 | 0.520 | 0.263 | 0.858 |

Notes: RES: resilient coping, INT: intrinsic goals, EXT: extrinsic goals, ECOL: ecological orientation, SOC: social orientation, ECON: economic orientation, SFC: sustainable fashion consumption.

**Table 4.** Assessment of discriminant validity: Heterotrait–Monotrait Ratio (HTMT).

|  | **RES** | **INT** | **EXT** | **ECOL** | **SOC** | **ECON** | **SFC** |
|---|---|---|---|---|---|---|---|
| RES | | | | | | | |
| INT | 0.639 | | | | | | |
| EXT | 0.439 | 0.629 | | | | | |
| ECOL | 0.264 | 0.477 | 0.086 | | | | |
| SOC | 0.215 | 0.400 | 0.096 | 0.712 | | | |
| ECON | 0.352 | 0.357 | 0.342 | 0.327 | 0.239 | | |
| SFC | 0.243 | 0.452 | 0.109 | 0.855 | 0.566 | 0.288 | |

Notes: RES: resilient coping, INT: intrinsic goals, EXT: extrinsic goals, ECOL: ecological orientation, SOC: social orientation, ECON: economic orientation, SFC: sustainable fashion consumption.

**Table 5.** Structural model results (N = 503).

| Relationship | ß | | t | Confidence Interval (95%) | VIF | $f^2$ |
|---|---|---|---|---|---|---|
| H1: Resilient Coping → Intrinsic Goals | 0.479 | *** | 11.690 | [0.398: 0.558] | 1.000 | 0.298 |
| H2: Resilient Coping → Extrinsic Goals | 0.352 | *** | 7.273 | [0.253: 0.447] | 1.000 | 0.141 |
| H3a: Intrinsic Goals → Ecological Orientation | 0.408 | *** | 8.105 | [0.309: 0.507] | 1.460 | 0.138 |
| H3b: Intrinsic Goals → Social Orientation | 0.429 | *** | 9.794 | [0.344: 0.515] | 1.242 | 0.175 |
| H3c: Intrinsic Goals → Economic Orientation | 0.124 | * | 1.989 | [0.002: 0.247] | 1.460 | 0.012 |
| H4a: Extrinsic Goals → Ecological Orientation | −0.139 | *** | 3.631 | [−0.214: −0.065] | 1.283 | 0.018 |
| H4b: Extrinsic Goals → Social Orientation | −0.253 | *** | 6.389 | [−0.331: −0.173] | 1.242 | 0.061 |
| H4c: Extrinsic Goals → Economic Orientation | 0.158 | *** | 3.505 | [0.067: 0.247] | 1.283 | 0.023 |
| H5a: Resilient Coping → Ecological Orientation | 0.098 | | 1.911 | [−0.004: 0.195] | 1.341 | 0.009 |

**Table 5.** *Cont.*

| Relationship | ß | | t | Confidence Interval (95%) | VIF | f$^2$ |
|---|---|---|---|---|---|---|
| H5b: Resilient Coping → Social Orientation | 0.102 | * | 2.245 | [0.017: 0.189] | 1.341 | 0.009 |
| H5c: Resilient Coping → Economic Orientation | 0.191 | ** | 3.272 | [0.075: 0.306] | 1.341 | 0.032 |
| H6a: Ecological Orientation → Sustainable Fashion Consumption | 0.777 | *** | 17.614 | [0.683: 0.858] | 1.850 | 0.858 |
| H6b: Social Orientation → Sustainable Fashion Consumption | 0.003 | | 0.050 | [−0.094: 0.104] | 1.769 | 0.000 |
| H6c: Economic Orientation → Sustainable Fashion Consumption | 0.028 | | 0.792 | [−0.040: 0.098] | 1.102 | 0.002 |
| | R$^2$ | | Q$^2$ | | | |
| INT | 0.230 | | 0.135 | | | |
| EXT | 0.124 | | 0.078 | | | |
| ECOL | 0.174 | | 0.125 | | | |
| SOC | 0.152 | | 0.114 | | | |
| ECON | 0.138 | | 0.102 | | | |
| SFC | 0.620 | | 0.450 | | | |

Notes: INT: intrinsic goals, EXT: extrinsic goals, ECOL: ecological orientation, SOC: social orientation, ECON: economic orientation, SFC: sustainable fashion consumption; significant at * $p < 0.05$, ** $p < 0.01$, *** $p < 0.001$.

*4.2. Structural Model Results*

The path analysis was conducted using partial least squares structural equation modeling (PLS-SEM) and tested the proposed hypotheses (see Figure 1). The path coefficients confirmed that resilient coping (RES) positively influenced both intrinsic goals (INT) and extrinsic goals (EXT), supporting H1 and H2 (H1: β = 0.48, $p < 0.001$, H2: β = 0.35, $p < 0.001$). Regarding the relationships between the intrinsic goals (INT) and the three dimensions of sustainable consumption orientation, consumers' INT had a significant positive effect on all three aspects—ecological (ECOL), social (SOC), and economic (ECON)—confirming H3 (H3a: β = 0.41, $p < 0.001$, H3b: β = 0.43, $p < 0.001$, H3c: β = 0.12, $p < 0.05$). As for the paths from extrinsic goals (EXT) to sustainable consumption orientation, the results differed from those obtained for INT. While EXT had a significant negative effect on two dimensions, ECOL and SOC, it had a positive effect on ECON, accepting H4 (H4a: β = −0.14, $p < 0.001$, H4b: β = −0.25, $p < 0.001$, H4c: β = 0.16, $p < 0.001$). We tested the direct relationship between resilient coping and sustainable consumption orientations, and there were significant positive direct effects of RES on SOC and ECON, but no statistically significant relationship between RES and ECOL, which indicated that H5 was partially accepted (H5b: β = 0.10, $p < 0.05$, H5c: β = 0.19, $p < 0.01$). Finally, we tested the relationships between the three dimensions of sustainable consumption orientation (ECOL, SOC, and ECON) and sustainable fashion consumption behavior (SFC); we found a significant positive path from ECOL to SFC, but not the other two paths, which suggested that H6 was partially accepted (H6a: β = 0.78, $p < 0.001$).

**5. Discussion**

This present study aimed to investigate the fundamental mechanisms behind consumers' sustainable consumption behaviors, focusing specifically on two crucial aspects: the adaptive coping strategies that consumers have developed in response to the COVID-19 pandemic and the evolution of their life goals, encompassing both intrinsic and extrinsic dimensions, as influenced by the pandemic.

This study reveals that consumers' resilient coping significantly influences both their intrinsic and extrinsic goals. The findings suggest that the methods individuals use and the skills they developed to navigate the pandemic-related challenges have reshaped their life goals and influenced their priorities in life values. This reshaping of goals and values highlights a transformative aspect of resilience, adding new insights into how adversity can lead to the reevaluation and strengthening of personal and social values. In line with previous research [14,20], it was observed that consumers, when confronted with

challenging circumstances, tend to increase their focus on self-awareness and concern for others. The results of this study offer insights that are pivotal for the fashion industry. For example, resilience shapes consumer behavior, prompting a shift toward more conscious, value-driven fashion choices. This shift is likely to influence fashion consumption trends, steering them toward environmentally conscious consumerism. It offers a novel perspective for the fashion industry, suggesting that resilient coping mechanisms not only shape consumer behavior but also drive a shift toward more conscious choices toward fashion products produced in an environmentally friendly manner, potentially influencing sustainable consumerism in the fashion sector.

Concerning the positive association of consumers' intrinsic goals with three aspects—ecological, social, and economic—of sustainable orientation, consistent with previous studies [23,51], it implies that personal intrinsic values related to self-acceptance, intimacy, and societal contribution are vital to sustainable orientation. As intrinsic goals are related to fundamental psychological needs (i.e., autonomy, competence, and relatedness), consumers who have established their intrinsic goals are more likely to pay their attention to sustainable consumption, and it has extended to the community and society levels where they are involved: environmental issues and the well-being of workers who are involved in the fashion industry, as well as companies' sustainable management and survival during the pandemic.

Importantly, when examining the influence of extrinsic goals on various aspects of sustainable consumption orientations, this study revealed a negative association between consumers' extrinsic goals and ecological and social sustainable orientations. This result aligns with previous studies indicating that extrinsic goals, often linked with materialism and status-seeking, tend to conflict with values of environmental preservation and social welfare [51,52]. These goals can overshadow the intrinsic motivation required for ecological and social responsibility, as they focus more on personal gain than on environmental well-being. Conversely, our findings revealed a positive association between extrinsic goals and economic sustainable orientation. This could be explained by the specific context of fashion consumption behaviors. In the fashion industry, economic sustainability often correlates with personal benefits such as cost savings and long-term value, which might align with extrinsic goals. Thus, consumers may perceive economic sustainability in fashion as an alignment with their personal financial interests, such as investing in durable or timeless pieces that offer longer-term value over fast-fashion, cheaper items. This perspective might motivate their positive association with economic sustainability, even as they deprioritize ecological and social aspects due to their extrinsic orientations.

As for the direct effects of resilient coping on sustainable orientation, this study confirmed that resilient coping positively correlates with social and economic orientations, while it does not show a significant relationship with ecological sustainable orientation. For instance, previous researchers have indicated that in times of the COVID-19 pandemic, individuals tended to prioritize immediate, tangible aspects of well-being, such as financial stability and equitable treatment in the workplace [5]. This aligns with the observed emphasis on social and economic orientations, as fashion consumers prioritized aspects like fair employment practices and long-term corporate success, which are perceived as directly impacting their personal livelihood. In contrast, the lack of a significant relationship between resilient coping and ecological sustainable orientation in the context of fashion consumption may be attributed to a more immediate focus on survival and financial sustainability. This offers new insights into the field by highlighting how resilience coping navigates sustainable orientation, particularly after the COVID-19 pandemic. It also emphasizes how consumer goals, particularly intrinsic ones, might serve as mediators in the subtle impact of resilient coping on ecological sustainable orientation. While resilient coping may not have a direct effect on ecological orientations, it underscores the vital role of consumers' inherent motivations in promoting environmental values within the fashion industry [21,23].

Indeed, the connection between a resilient mindset and ecological orientation through intrinsic goals is crucial, especially in light of this study's findings. Our finding indicates

that prioritizing care for the ecological environment is a driving force for sustainable fashion consumption. These findings suggest that consumer goals, reshaped by resilient coping during the pandemic, positively influence ecological orientation, which in turn is a key driver of sustainable fashion consumption. Also, the positive association between the ecological orientation and sustainable behavior of fashion consumers, supported by previous research, is reaffirmed. Previous scholars underscored the impact of consumer optimism and pessimism on green apparel purchase behavior [67], and our study extends those insights to the realm of ecological consciousness in driving sustainable fashion consumption. While previous research indicated that environmental orientation mediates the relationship between consumer attitudes and green apparel purchasing, our findings reveal strong support for ecological orientation in sustainable fashion consumption. However, social and economic aspects of sustainability orientation show less significance. A plausible explanation is that the general public may lack a comprehensive understanding of sustainability's broader dimensions, especially the social and economic aspects, in contrast to the ecological aspects, which are more straightforwardly linked to environmental sustainability. This observation opens up potential scholarly avenues for further investigation, particularly in enhancing public awareness and understanding of the full spectrum of sustainability. As our study centers on sustainable fashion consumption, particularly highlighting choices driven by the environmental impacts of fashion business practices, it logically emphasizes ecological orientation over social and economic orientations. However, given the lesser impact of social and economic orientation observed in the results, future scholars have the opportunity to broaden the scope of sustainable fashion consumption to encompass consumer choices that consider the ethical and social justice aspects of fashion business practices. Such an expansion could stimulate scholarly debates that reshape the discourse surrounding our study more comprehensively.

## 6. Conclusions

### 6.1. Theoretical Implications

This study integrates Goal Content Theory (GCT) with the Consumer Sustainability Orientation (CSO) framework, which provides theoretical implications. Firstly, this study extends the application of GCT to sustainable consumption. By examining both intrinsic and extrinsic goals within sustainable fashion consumption, this study offers insight into the relationships between different types of goal motivations and consumer sustainable consumption behavior. Secondly, by adding GCT into the CSO framework, this study found that consumption behaviors are not just influenced by a general orientation toward sustainability but are also deeply rooted in individual psychological motivations and goals. This extended framework enriches the existing CSO framework, allowing for the explanation of the complexities of sustainable consumer behaviors in the fashion industry in a comprehensive manner. Moreover, this study contributes to the fashion academic field by portraying how changes brought by the pandemic have significantly shifted consumer priorities, values, and behaviors, particularly in the post-pandemic era. This study's findings point out that the COVID-19 pandemic not only affects consumer goals and motivations but also has an impact on reshaping their sustainability orientations and fashion consumption practices that are closely associated with social and environmental issues. Finally, this study bridges the gap between the psychological theoretical framework and practical consumer behavior in the context of sustainability. It provides a holistic view of the psychological mechanisms of sustainability-oriented consumer choices, particularly in fashion consumption.

### 6.2. Managerial Implications

This study offers valuable practical and managerial implications. Firstly, fashion companies need to deeply integrate sustainability and ethical practices into their business strategies, aligning with consumers' evolving intrinsic values and resilience-driven behavior. For example, fashion brands can implement initiatives that focus on the well-being

of workers in their supply chain, ensuring fair labor practices and fostering community development. Fashion retailers can emphasize transparent communication with consumers, providing viable and feasible long-term plans to achieve sustainable business goals even with unpredictable, harsh market situations in the future. These efforts for public communication would demonstrate their commitment not only to environmental sustainability but also to social responsibility and economic resilience.

We also found different roles of intrinsic and extrinsic goals in explaining and supporting fashion consumers' sustainable orientation. Considering the differential impact of extrinsic and intrinsic goals on sustainable consumption orientations, there is the need to strategically balance and tailor marketing and product strategies to cater to these diverse consumer motivations. For consumers with intrinsic goals, fashion companies should focus on promoting the ethical and environmental aspects of their products. This could involve marketing campaigns that emphasize the positive impacts of purchasing such products on the environment and society, appealing to the intrinsic values of self-acceptance, intimacy, and societal contribution. Conversely, for consumers with extrinsic goals, who show a positive association with economic sustainable orientation, fashion companies could emphasize the long-term economic benefits associated with their products and services by showcasing to consumers long-term investments over cheaper and/or fast-fashion alternatives. Highlighting sustainable aspects focusing on long-lasting quality could lessen consumers' concerns regarding fashion companies' sustainable efforts toward ecological and social dimensions, which eventually connect to their choices of sustainable fashion products. This balanced strategy allows fashion brands to effectively cater to a broader market segment while promoting sustainable fashion consumption.

Finally, the findings of this study highlight the importance of policy intervention in promoting sustainable fashion consumption. Policymakers in the fashion industry could leverage these insights to implement policies and regulations (e.g., regulating fast fashion) that foster environmental consciousness among consumers. This would encourage sustainable and environmentally conscious business practices among fashion brands and companies. For instance, industry-wide sustainability standards could require transparency in the manufacturing and sourcing processes, especially when fashion companies subcontract with vendors or manufacturers in developing countries. This would ensure consumers have the information necessary to make informed decisions about their fashion consumption.

*6.3. Limitations and Future Studies*

While this study offers valuable insights, there are some limitations. Firstly, while the proposed research framework is well constructed based on theoretical foundations, there might be additional factors that could be considered in our model. Future studies could explore uncharted variables such as psychological traits (i.e., self-identity, motivations, attitude) and cultural factors (i.e., social norms) to provide a more comprehensive understanding of resilient coping and its influence on sustainable orientation and fashion consumption. Secondly, expanding the scope of participant samples by including more diverse populations will enhance the generalizability of findings. This broader perspective can help uncover insights into the connections between resilient coping, sustainable orientations, and sustainable fashion consumption across various demographic groups (i.e., gender and income). Thirdly, sustainable fashion consumption primarily addresses ecological measures. Consequently, the findings suggest that ecological orientation emerged as the primary influencer that directly explains sustainable fashion consumption. While this association is logical, the fact that social orientation does not influence sustainable fashion consumption implies that consumers may distinguish environmental concerns (i.e., harmful effects on the environment) from ethical concerns (i.e., fair labor issues in the fashion industry). This leaves opportunities for future research to explore how consumer choices and consumption behaviors may vary based on ethical and holistic views of sustainable orientation toward the fashion industry and its associated concerns and highlight the need to

explore the nuanced interactions among social, economic, and ecological factors in shaping consumer behavior on sustainable fashion consumption. It also implies that in the fashion industry, retailers' social and economic aspects of sustainable business practices need to be aligned with consumers' sustainable orientations. Furthermore, although this study offers valuable insights into sustainable fashion consumer behavior during the COVID-19 pandemic, our findings cannot generalize the complex nature of consumer behaviors and reactions. Thus, future researchers should acknowledge the diversity in consumer responses to the pandemic, recognizing that not all individuals responded to the pandemic in a similar manner. Finally, this study arises from the online survey methodology employed. This approach may introduce response bias or limit the depth of information gathered due to the absence of face-to-face interactions. Future research might consider different methodological approaches, such as mixed-method approaches or in-person interviews, to address these potential limitations. Also, conducting longitudinal studies could capture a more holistic view, especially regarding changes in environmental and societal conditions.

**Author Contributions:** Conceptualization, J.H., S.-y.Y. and L.Z.; data curation: J.H. and S.-y.Y.; data analysis: J.H. and S.-y.Y.; writing—original draft preparation, J.H., S.-y.Y. and X.S.; writing—review and editing, J.H., S.-y.Y., X.S. and L.Z.; project administration and supervision, J.H. and S.-y.Y.; funding acquisition, L.Z. All authors have read and agreed to the published version of the manuscript.

**Funding:** This research received no external funding.

**Institutional Review Board Statement:** The study was conducted in accordance with the Declaration of Helsinki and approved by the Institutional Review Board of University of Missouri-Columbia (IRB project number: 2062303, date of approval: 23 June 2021).

**Informed Consent Statement:** Informed consent was obtained from all subjects involved in the study.

**Data Availability Statement:** The data presented in this study is available and can be provided by the authors upon a reasonable request.

**Conflicts of Interest:** The authors declare no conflict of interest.

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
