# Peer review of "Sustainable Fashion in New Era: Exploring Consumer Resilience and Goals in the Post-Pandemic"

_sustainability, doi:10.3390/su16083140_

Round 1
Reviewer 1 Report
Comments and Suggestions for Authors
This study adds value to a growing body of research. However, minor edits for clarity are needed.
New and novel concept:
The framework being tested is compelling and relevant to current considerations in sustainable consumer behavior.
Introduction/Literature:
While the term "Fashion" is frequently used in the introduction and conclusion, it is not clear how this context was established for study participants nor why the fashion industry was explored. Please provide some contextualization.
I will address this further in the methods section, but I suggest using additional language to introduce the ecological component of SFC. This will set the stage of the tested relationship to environmental/ecological considerations only.
Theory:
There are several theories and frameworks (GCT, SDT, CSO) introduced in the abstract and literature review. The quantity makes it a bit difficult to distinguish your perspectives and assumptions. Explain which are assumptions and which frame the study. For instance, SDT may be introduced when describing GCT, but does not appear to be the foundation of your study. Similarly, TBL may be useful background to CSO, but ensure it is clear that your study uses GCT as theoretical basis and CSO to actualize application to sustainable fashion consumption.
Methods:
The introduction states this is an exploration of fashion consumers, but the context to ensure the MTurk participants were considering their fashion purchases as not been made clear. Was the only point when fashion was referenced in SFC? Were participants prompted to consider their fashion purchasing habits when starting the survey? Provide context and clarity.
Intrinsic and extrinsic motivation statements are not phrased in the same manner as other statements. Was there a starting phrase to indicate participant interpretation? If so, it would add clarity to see the consistency in the participants prompts. For instance, there is not a clear "Agree/Disagree" to "Self-acceptance/personal growth: Being happy and having a very meaningful life." But this may be more clear if we knew the starting phrase such as "When shopping for fashion products, I prioritize...".
The measured items for sustainable fashion consumption (SFC) is out of line with the cited literature and the introduction of the concept in the literature review. The measured items only address Ecologically Sustainable Fashion Consumption (ESFC for ease of writing), ethical considerations are not accounted for.
Results:
Your tables and figures are easy to follow and convey results well.
Discussion and Conclusion:
Due to SFC only addressing ecological measures, there are also gaps in clarity for the discussion and future research opportunities. For instance, the fact that ECOL affects ESFC is logical, the fact that SOC does not affect ESFC may also be more clearly explained with participants distinguishing environmental and ethical actions. This then leaves future research opportunities to look at ethical and holistic SFC.
The following statement distinguishes ethical and sustainable, but in your literature review you address ethics as a part of sustainability. This should be revisited for clarity and consistency. Statement: "The results of this study offer insights that are pivotal for the fashion industry. For example, resilience shapes consumer behavior, prompting a shift toward more conscious, value-driven fashion choices. This shift is likely to influence fashion trends, steering them toward sustainability and ethical consumerism. It offers a novel perspective for the fashion industry, suggesting that resilient coping mechanisms not only shape consumer behavior but also drive a shift toward more conscious and value-driven fashion choices, potentially influencing trends toward sustainability and ethical consumerism in the fashion sector."
Additional considerations in that statement: use of language should be refined, for instance, "fashion trends" seems to actually refer to "consumption trends". Additionally, significance was not found for H5a, H6b, or H6c which seems out of alignment with the conclusions being presented.
References:
References are holistic and representative of recent research and current perspectives.
Reviewer 2 Report
Comments and Suggestions for Authors
1.This study makes a significant contribution to the current understanding of how COVID-19 has influenced consumer behavior, particularly in the context of sustainable fashion consumption. The use of Goal Content Theory (GCT) and Consumer Sustainability Orientation (CSO) framework is a novel approach that promises to offer substantial insights. However, a few concerns and areas for improvement are discussed below.
(1) Although this study provides helpful insights into the behavior of sustainable fashion consumers during the COVID-19 pandemic, it may overgeneralize the nature of consumer behavior. It is essential to consider that not all consumers responded to the pandemic in the same manner. Future research should also consider this aspect.
(2)The introduction discusses the restructuring of fashion companies and trends toward online purchasing without citing specific empirical evidence to support these claims. More concrete data could strengthen these arguments and provide a clearer picture of the impact of the pandemic on fashion companies and consumer behavior.
(3)The paper could do a better job of precisely defining what is meant by “sustainable fashion.” Is it ethical sourcing? Is there low-environmental-impact production? Alternatively, is the longevity of the product.
(4)The introduction was better structured, and some arguments were somewhat repetitive. It is highly recommended to go straight to this point and avoid unnecessary repetitions.
2.While the literature review section of this paper offers a comprehensive exploration of resilient coping, Goal Content Theory, Consumer Sustainability Orientation, and the relationships among these concepts, some gaps and areas for revision have been identified.
(1) This review could delve more into the current state of knowledge and existing debates or contrasting views in the field. It mainly reports widely accepted principles, but it is also crucial to indicate divergent theories or unsolved questions that exist in this field of study.
(2)The terms’ resilient coping" and "sustainable fashion consumption" are used extensively throughout the literature review, but these terms are not given concise definitions. The review could benefit from clear and succinct definitions of these terms earlier in the text.
3.The discussion section of the study provides a thoughtful reflection on the results and their implications, albeit with some areas of potential enhancement.
(1)While the study implies the consequences of the findings, a more explicit discussion of the larger implications, particularly for policymakers or the fashion industry, would further strengthen the paper.
(2)Certain points appear twice, such as remarks on resilient coping mechanisms shaping behavior and influencing trends in the fashion industry. Greater attention to the flow of discussion could improve readability and cohesiveness.
4.The conclusion section of the study appears well-structured and provides comprehensive coverage of theoretical and practical implications, limitations, and directions for future research.
Reviewer 3 Report
Comments and Suggestions for Authors
The manuscript entitled “Sustainable Fashion in New Era: Exploring Consumer Resilience and Goals in the Post-Pandemic” deals with an interesting and important topic that fits perfectly in the journal’s scope. I have, however, some major and minor concerns that need to be addressed.
In line 33, the proper term is ultra-fast fashion. H1, H2, H3, and H4 seem to be obvious as you properly explain them in the literature review. I would recommend you to specify H1 and H2 (resilient coping with Covid-19, as you argue in the manuscript), this was they were not too general and explained by previous research; although this way your questionnaire items would not measure that specific coping. Please, refer to Figure 1 in the text. Please, specify the date of your data collection and provide a table containing the distribution of your sample based on demographic data beside gender and race (e.g., age, country, income, education level, settlement type). Some of the results of the structural model mentioned on p. 10 do not correspond with data in Table 6: based on the table,
- H1: beta = 0.48 instead of 0.47;
- H5b: beta = +0.10 and p < 0.05 instead of -0.1 and 0.01;
- H5c: p < 0.01 instead of 0.05.
In the conclusions, please, explain your statement: “This implies potential gaps between consumers' understanding of sustainability and fashion retailers' actual sustainable practices.” Finally, there are some formatting issues; see, e.g., the format of in-text citations and the list of references; missing heading numbering; H5 in line 281 is not bold; H1 in Table 4 is unnecessarily bold.
Comments on the Quality of English LanguageThere are some typos in the manuscript, see, e.g., lines 76, 328.
Round 2
Reviewer 2 Report
Comments and Suggestions for Authors
The authors have effectively addresses my comments and remarks and the quality and clarity of the manuscript have been improved. I can now recommend acceptance.